# The Physiological Cost of Being Hot: High Thermal Stress and Disturbance Decrease Energy Reserves in Dragonflies in the Wild

**DOI:** 10.3390/biology14080956

**Published:** 2025-07-29

**Authors:** Eduardo Ulises Castillo-Pérez, Angélica S. Ensaldo-Cárdenas, Catalina M. Suárez-Tovar, José D. Rivera-Duarte, Daniel González-Tokman, Alex Córdoba-Aguilar

**Affiliations:** 1Posgrado en Ciencias Biológicas, Universidad Nacional Autónoma de México, Av. Ciudad Universitaria 3000, Coyoacán, Mexico City 04510, Mexico; ulises.castillo@iecologia.unam.mx (E.U.C.-P.);; 2Instituto de Ecología, Universidad Nacional Autónoma de México, Circuito Exterior, Ciudad Universitaria, Coyoacán, Mexico City 04510, Mexico; 3Instituto de Investigaciones en Ecosistemas y Sustentabilidad, Universidad Nacional Autónoma de México, Morelia 58190, Mexico; csuarez@cieco.unam.mx; 4Laboratorio de Hidrobiología, Departamento de Ecología y Recursos Naturales, Escuela de Biología, Facultad de Ciencias, Universidad Nacional Autónoma de Honduras, Tegucigalpa 11101, M.D.C., Honduras; jdr495@hotmail.com; 5Red de Ecoetología, Instituto de Ecología A.C, Carretera Antigua a Coatepec 351, El Haya, Xalapa 91073, Mexico; daniel.gt@inecol.mx

**Keywords:** conservation, dragonflies, tropics, disturbance, body size, thermal stress, protein, lipid

## Abstract

Human activities can increase environmental temperatures and alter habitats, which may negatively affect insects—particularly those that experience strong differences between their body and environmental temperature, known as thermal stress. In this study, we found that dragonflies living in preserved dry forest sites showed higher thermal stress at lower maximum temperatures, while those in disturbed sites maintained consistent levels of thermal stress. Dragonflies under higher thermal stress tended to have lower amounts of lipids and proteins, which are key energy reserves. Individuals from disturbed sites also had lower energy reserves than those from preserved sites. We found a weak positive relationship between protein content and mean temperatures. Interestingly, individuals from preserved sites had larger thoracic mass, but this was only observed at high temperatures. Our results suggest that dragonflies exposed to both habitat disturbance and high thermal stress may be in poorer energetic condition and could be more vulnerable as temperatures continue to rise and natural habitats degrade.

## 1. Introduction

Anthropogenic disturbances, such as land-use changes for livestock farming, agriculture, and urban expansion, have altered environmental conditions, leading to water and air pollution and deforestation [1,2,3]. Reduction in vegetation cover modifies microclimatic conditions, typically leading to higher temperatures due to increased solar radiation exposure and reduced humidity [4,5,6]. These hotter and drier conditions can be detrimental or even lethal to ectothermic insects since they depend on environmental temperature to perform essential functions such as foraging, development, and reproduction [7,8,9,10]. In tropical regions, microclimatic variability and vegetation structure have been identified as important predictors of butterfly abundance and composition [11]. Similarly, in Odonata (dragonflies and damselflies), abiotic factors such as temperature—mediated by forest cover—act as environmental filters determining which species successfully colonize a given site [12]. Shifts in Odonata community composition in cloud forest have also been strongly associated with local variation in canopy and light availability [13]. The effects of microclimate on insect fitness are mediated by both behavior and physiology; thus, microclimatic variation plays a key role in determining insect distribution at both local and broader scales [14,15,16]. For example, Bota-Sierra et al. [17] demonstrated that microhabitat in tropical odonates is linked to their thermal tolerances. In addition, Odonata populations inhabiting disturbed habitats in tropical dry forests—where vegetation cover is reduced and local temperatures are higher—have been shown to exhibit increased thermal tolerance [18].

Temperature increase and habitat disturbance have been linked to global declines in insect abundance and diversity [3,16,19]. However, some species persist in disturbed environments, suggesting that certain functional traits enhance their thermal tolerance. A central concept in this context is thermal stress, defined as the difference between body temperature and ambient temperature. Body temperature is influenced by both environmental conditions and internal factors, such as muscular activity before and during flight [20,21,22]; consequently, thermal stress increases when body temperature rises well above ambient levels. Higher thermal stress values can reflect greater physiological demand to maintain optimal function, especially under elevated environmental temperatures [23]. This demand may limit the ability of some species to persist in disturbed environments unless they possess traits that mitigate such stress. For example, Giménez Gómez et al. [20] showed that smaller dung beetles with lower thermal stress could exploit disturbed open-canopy habitats more effectively. Similarly, Rocha et al. [24] reported that heliothermic odonates—species that actively absorb solar radiation to elevate their body temperature—are more frequently found in disturbed habitats with greater sun exposure, whereas thermoconforming species—those whose body temperature closely follows ambient temperature—tend to occur in preserved, shaded habitats.

Species from the suborder Anisoptera (also called dragonflies, order Odonata) have been described as capable of tolerating extreme temperature conditions and highly disturbed habitats. Anisopterans are typically found flying in open, sunny–exposed environments, which they can exploit due to their larger body size, high biomass, and superior thermoregulatory capacity [25,26]. For instance, Castillo-Pérez et al. [27] found a positive correlation between dragonfly taxonomic diversity and higher temperatures, while Suárez-Tovar et al. [28] reported that dragonfly communities were relatively unaffected by urbanization. This tolerance has been associated with dragonflies’ high dispersal ability and their thermoregulatory capacity, which relies on both behavioral and physiological mechanisms [25,29,30]. Some of the behavioral mechanisms observed in dragonflies include the obelisk position, where individuals modify the angle of the abdomen to minimize the surface area exposed to direct sunlight [25]. Dragonflies may also seek shaded microhabitats or adjust their flight patterns to avoid overheating [26]. On the physiological side, certain species can regulate thoracic temperature during flight by altering hemolymph circulation from the thorax to the abdomen [31,32]. This redirection allows excess heat produced by flight muscles to be dissipated via the abdomen, which functions as a heat sink [33]. Such physiological and behavioral thermoregulation enables dragonflies to maintain optimal muscle performance under high ambient temperatures. However, sustaining these strategies entails a metabolic cost: as body temperature rises, metabolic rate increases accordingly, elevating the energy required to sustain both flight and thermoregulation [34,35]. When environmental conditions limit resource availability, this demand may impose a significant energetic burden [36]. In such cases, dragonflies often rely more heavily on behavioral thermoregulatory strategies—such as seeking shaded microhabitats or modifying flight activity—to further avoid overheating [34,35,37]. These energetic costs may constrain foraging and mating opportunities, eventually leading to a depletion of energy reserves [38]. For instance, a substantial body fat reduction can be caused by the death of fat body cells, as found in *Drosophila melanogaster* [39]. The death of such cells reduces the ability to store more lipids.

Dragonflies inhabiting tropical dry forests can experience high thermal stress in disturbed sites, due to high temperatures, reduced vegetation cover, and periods of water scarcity [18,40]. Besides modifying environmental temperature, habitat disturbance can reduce prey availability and lower nutritional quality [41], which may further impact energy reserves and body size of individuals in disturbed environments. This reduction in energy availability may, in turn, increase the energy costs of thermoregulation, although the underlying mechanisms are not yet fully understood.

Body size has also been identified as a trait influencing thermal stress, as smaller odonates are capable of maintaining lower basal metabolic rates (BMRs) under high-temperature stress compared to larger individuals [42]. As indicated before, high temperatures and thermal stress may lead to a decline in energy reserves [36,39]. Therefore, the lower BMR in smaller dragonflies potentially reduces their energetic demands in extreme heat conditions, providing an advantage in disturbed environments where resource availability is limited. On the other hand, some studies have reported that larger species may exhibit higher maximum thermal limits [15], potentially due to a greater water storage capacity [43]. Therefore, while smaller body size could reduce energetic costs under heat stress, larger size might enhance thermal tolerance. The extent to which dragonflies can modulate thermal stress may depend on habitat conditions, with individuals in preserved and disturbed environments showing different behavioral thermoregulatory responses.

This study investigated the relationship between thermal stress and the energetic condition of dragonflies in preserved and disturbed sites in a highly degraded and fragile tropical dry forest with high temperatures and a pronounced dry season in western Mexico. Due to the difficulty of observing and identifying females in the field, this study was limited to males. Accordingly, the following predictions apply specifically to males. First, we predicted that individuals who inhabit preserved sites with lower temperatures will show lower levels of thermal stress, while individuals who inhabit disturbed sites with higher temperatures will exhibit higher thermal stress. Second, we predicted that individuals from disturbed sites will have a smaller thoracic mass, reflecting possible reductions in structural investment under resource-limited, high-temperature conditions. Third, we expect that individuals with higher levels of thermal stress—especially larger-bodied individuals with higher energetic demands—will show poorer energetic reserves (i.e., lipid or protein reserves). In contrast, smaller individuals may be less affected by thermal stress due to lower metabolic requirements.

## 2. Materials and Methods

### 2.1. Study Site

We carried out this study in the deciduous tropical forest of the Chamela-Cuixmala Reserve and its surroundings, located in the municipality of La Huerta, Jalisco, Mexico (19°29′39.8″ N, 105°02′48.4″ W). The protected area of the Chamela-Cuixmala Reserve covers 13,142 ha, while the surrounding landscape, where the disturbed sites are located, encompasses approximately 191,158 ha [44]. The area features a warm, subhumid climate with a pronounced dry season from November to June [45]. The average annual rainfall is 832  ±  277 mm, though there is wide variation between years, ranging from 340 to 1394 mm [46]. During the dry season (November–June), temperatures range from 8 °C to 38 °C, while in the rainy season (July–October), they range from 16 °C to 39 °C [44,47]. The area comprises a mosaic of intact tropical forests, agricultural and livestock fields, and second-growth forests in multiple successional stages [48,49].

Data were collected in 14 sites: 9 were anthropogenically disturbed, and 5 were preserved (Figure 1). Disturbed sites consisted of ponds, rivers, or streams outside the Chamela-Cuixmala Reserve, on non-conservation land, near houses, hotel complexes, agricultural plots, and cattle pastures [44]. In contrast, preserved sites included similar water bodies—ponds, rivers, or streams located within the Chamela-Cuixmala Reserve, where forests have been under protection for over 50 years and represent an important remnant of native old-growth vegetation [46]. Furthermore, the preserved sites in this study have a higher percentage of vegetation cover compared to the disturbed sites [44,48].

### 2.2. Environmental and Body Temperature Recording

Between 6 September and 5 October 2019, and between 31 August and 30 September 2022, we recorded ambient temperature data at 14 sampling sites using HOBO^®^ data loggers (MX 2201, ±0.5 °C). Data loggers were placed 1.5 m above the ground on tree branches, exposed to sunlight, and approximately 2 m from the water body. Due to logistical and budgetary constraints, data loggers were deployed for a limited duration, recording temperatures every five minutes over an average of five full non-consecutive days per site (with a minimum of two full days). From these measurements, we calculated mean and maximum ambient temperatures for each site. The number of recording days, as well as the resulting temperature values, are detailed in Appendix A.

The body temperature of adult male anisopterans was recorded between 10:00 and 15:00 h (UTC-6), which corresponds to the hours of maximum activity in odonates. To determine the individuals’ body temperatures, we captured them using an entomological net while they were flying. After capture, we held individuals by the wings with a set of metal forceps and took three thermographic images at 3 s intervals using a FLIR^®^ model E6 camera (resolution of 160 × 120 pixels with a spectral range of 7.5–13 µm and a thermal sensitivity of <60 mK at 30 °C, and accuracy of ±2 °C). The camera was calibrated using the standard calibration service provided by FLIR^®^ to guarantee accuracy. Heat exchange between the observer and the experimental individual was reduced by taking the thermographic images within five seconds of gripping the wings with the forceps, avoiding exposing individuals to direct solar radiation when the thermographic image was taken. Once the thermographic images were stored in the memory of the thermographic camera, the right hindwing of each individual was marked with a fine-tipped black marker to prevent recapture. Additionally, we measured total body length, extending from the head to the last abdominal segment, as a measure of body size [50] using a Lion Tools digital caliper (±0.05 mm). Individuals were released at the same site where they were captured.

### 2.3. Thermographic Image Analysis and Thermal Stress Analysis

We used FLIR Tools^®^ version 6.4 software to obtain temperatures of the synthorax (1 mm of perimeter), which is the warmest tagma of odonates [30]. The average of the three thermographic measurements taken for each individual was considered the body temperature (*T_b_*). To estimate thermal stress, we followed the method described by Giménez Gómez et al. [20], which consists of calculating the difference between body temperature and the ambient temperature (*T_b_* − *T_a_*) at the time of capture, by subtracting ambient temperature from the average thoracic temperature.

### 2.4. Energy Reserves Calculation

We captured males of the seven most common anisopteran species—occurring in both preserved and disturbed sites—using an entomological net between 31 August and 30 September 2022, from 10:00 to 15:00 h (UTC-6). Only individuals from these seven species were used for body condition analyses (Appendix A). After capture, individuals were placed in centrifuge tubes and stored in a cooler at 5 °C to reduce the metabolic rate of dragonflies and prevent further activity, for a maximum of five hours before being transferred to a −20 °C freezer. Subsequently, they were sacrificed in a freezer at −20 °C, where they remained until we performed the individual condition analyses.

Protein and lipid concentrations (µg/mg) were quantified for each individual using the protocol of Foray et al. [51]. We used the thorax of individuals, without appendages, because it contains the flight muscles and biochemical reserves used in energetically demanding activities such as flying [52]. Lipids, in particular, exhibit a slower turnover rate in insects and are less sensitive to short-term post-capture delays [53,54]. The thorax was isolated and its mass was measured using a Velab VE-210 analytical balance (sensitivity: 0.1 mg; maximum capacity: 210 g). To prepare the samples, thoraxes were placed in 2 mL centrifuge tubes with steel beads and 180 µL of phosphate lysis buffer (100 mM KH_2_PO_4_, 1 mM DTT, 1 mM EDTA, pH 7.4). The tissues were homogenized into a hyaline solution using a TissueLyser II bead mill (Qiagen, Valencia, CA, USA). Subsequently, we took 180 µg of the sample and put it into a new tube to be centrifuged at 180× *g* relative centrifugal force (RCF) for 5 min at 4 °C. Next, we collected 2.5 µg of the supernatant in duplicate and placed it into a 96-well microplate. Then, 250 µg of Bradford reagent was added to each well. The plates were incubated for 20 min, and the absorbance was measured spectrophotometrically at 595 nm using an EL × 800 spectrophotometer (BioTek, Winooski, VT, USA). Protein content in the tissue was determined by comparing absorbance values with a calibration curve prepared using a dilution series of bovine serum albumin.

We used Van Handel’s [55] method, modified by Foray et al. [51] for lipid measurement. From the remaining 175 µL after taking the aliquots for the protein assay, we added 5 μL of phosphate lysis buffer and 1500 μL of methanol: chloroform solution (2:1). The mixture was vortexed for 2 min and centrifuged twice for 15 min at 180× *g* relative centrifugal force (RCF). Then, 100 μL of the supernatant was transferred in duplicate to a U-bottom plate and heated at 90 °C in a water bath until complete evaporation. After evaporation, 10 μL of 98% sulfuric acid was added to each well, and the plate was incubated at 90 °C for 2 min. The reaction was cooled, followed by the addition of 190 μL vanillin reagent prepared on the same day of use, with vanillin and phosphoric acid (68%) at a concentration of 1.2 g/L. After a 15 min incubation period, absorbance was measured at 515 nm. Lipid content was calculated using a calibration curve based on a glyceryl trioleate lipid dilution series.

### 2.5. Statistical Analysis

We employed linear mixed-effects models (LMMs) to assess the effects of maximum temperature, mean site temperature, site condition (preserved/disturbed), and body size on thermal stress. Fixed effects included site condition (preserved/disturbed), body size, maximum temperature, and the interaction between site condition and maximum temperature. Collection year, sampling site, and species identity were added as random effects. However, the sampling site explained a negligible proportion of the variance and was excluded to simplify the model. Maximum temperature was retained as a fixed effect instead of mean temperature because it improved the global model fit.

To analyze the relationship between energetic reserves (protein and lipid content) and thoracic mass with mean thermal stress, site condition, and both mean and maximum site temperatures, we constructed separated linear mixed-effects models (LMMs) for each response variable (protein content, lipid content, and thoracic mass). Species identity and sampling site were initially included as random effects, but the inclusion of the sampling site was evaluated individually for each response variable. For lipid and protein content, site accounted for a negligible proportion of variance and did not improve model support; it was therefore excluded for parsimony. In contrast, for thoracic mass, including site as a random effect improved model support and was thus retained. Mean site temperature was retained in all models of condition and thoracic mass was retained as a fixed effect instead of maximum temperature because it improved the global model fit. All response variables were log-transformed to meet model assumptions of normality and homogeneity of variance.

Prior to model selection, we assessed multicollinearity among predictors using the variance inflation factor (VIF) and found no indication of collinearity (VIF < 5). Candidate models were compared using the Akaike Information Criterion corrected for small sample size (AICc) following Burnham and Anderson [56]. Additionally, we calculated Akaike weights (w) to quantify the relative support for each model. A difference in AICc greater than 2 was considered evidence of weaker support for the competing model relative to the best-supported model. When two or more models had ΔAICc < 2, model averaging was used to account for model uncertainty. We report model-averaged estimates ± 95% confidence intervals.

Model validation for the linear mixed analyses was conducted visually through diagnostic plots, including theoretical quantiles versus standardized residuals (normal Q-Q plot) and residuals versus fitted values. Additionally, Shapiro–Wilk tests were performed to assess residual normality, and Breusch–Pagan tests were used to check for homogeneity of variance using the “performance” package version 0.11.0 [57]. All analyses were performed using R version 4.3.2 [58] using the lme4 package version 1.1-37 [59], and model-averaged predictions were computed using the AICcmodavg package version 4.3.2 [60].

## 3. Results

### 3.1. Environmental Temperatures

Disturbed sites showed higher air temperatures (35.71 °C [±4.02 °C SD]) compared to preserved sites (32.60 °C [±4.35 °C SD]). Although mean maximum temperatures were similar between disturbed and preserved sites, the three sites with the highest recorded maximum temperatures were all recorded in disturbed sites: La Meza (49.63 °C), Francisco Villa (49.42 °C), and Xametla (47.83 °C) (Appendix A).

### 3.2. Thermal Stress

We analyzed thermal stress in 393 adult individuals belonging to 16 species (12 species in preserved sites and 14 in disturbed sites; Appendix A) during two sampling periods (2019 and 2022). Based on AICc model selection (Appendix A), two linear mixed-effects models received substantial support (ΔAICc < 2), and we therefore used model averaging to account for model uncertainty. The most strongly supported predictors included maximum temperature, site condition (preserved/disturbed), and their interaction, with weaker support for body size. This interaction indicated that thermal stress decreased more markedly with increasing maximum temperatures in preserved sites (Figure 2). Accordingly, individuals from preserved sites exhibited higher thermal stress than those in disturbed sites, but only under the lowest maximum temperatures—that is, at the lowest values within the observed range of maximum temperatures across sites. Body size showed only a weak and inconsistent positive association with thermal stress (Appendix A).

### 3.3. Energy Reserves

We assessed the energetic reserves (lipids and proteins) and thoracic mass exclusively in males of the seven most common anisopteran species, collected in 2022 from preserved (114 individuals) and disturbed (126 individuals) sites (Appendix A). Based on AICc model selection, thermal stress emerged as the strongest and most consistent predictor across all three response variables (Appendix A).

For protein content, the two best-supported models (ΔAICc < 2) indicated a negative association with thermal stress and lower values in individuals from disturbed sites (Appendix A; Figure 3A,B). Lipid content followed the same pattern, with the three best-supported models (ΔAICc < 2) showing decreases associated with both higher thermal stress and site disturbance (Appendix A; Figure 3C,D).

For thoracic mass, the two best-supported models (ΔAICc < 2) included an interaction between thermal stress and site condition (Appendix A). This interaction indicated that thoracic mass increased with thermal stress in both preserved and disturbed sites, but the slope was slightly steeper in preserved sites—that is, individuals from preserved sites tended to exhibit higher thoracic mass under elevated levels of thermal stress (Figure 3E). Mean temperature appeared only as a weak predictor across all models (Appendix A). However, it showed a small and consistent positive effect on protein content in the best-supported models, although this effect was only subtly reflected in the predicted values derived from model averaging (Appendix A). Its effect on thoracic mass and lipid content remained weak and inconsistent (Appendix A).

## 4. Discussion

The main salient results of our research are as follows. First, based on data from 16 dragonfly species collected in two periods (2019 and 2022), individuals from preserved sites experience higher thermal stress at lower maximum temperatures but can reduce their thermal stress at higher temperatures, whereas dragonflies from disturbed sites maintain a consistent level of thermal stress across different maximum temperatures. Second, using data exclusively from the seven most common anisopteran species collected in 2022, we found that individuals experiencing higher thermal stress and greater thoracic mass had lower protein and lipid content. Third, again focusing only on these seven most common species present in both preserved and disturbed sites, while thoracic mass was also positively associated with thermal stress, this relationship depended on site condition: individuals from preserved sites exhibited a stronger increase in thoracic mass with thermal stress compared to those from disturbed sites. Additionally, individuals from preserved sites exhibited higher energetic reserves (protein and lipid content) than those from disturbed sites. Finally, we found a weak but consistent positive relationship between mean ambient temperature and protein content.

Contrary to our predictions, individuals from preserved sites showed higher thermal stress at lower maximum temperatures; however, they reduced their thermal stress as maximum temperatures increased. In contrast, individuals from disturbed sites maintained a constant level of thermal stress at all maximum temperatures. We hypothesize that individuals from disturbed sites may be engaging in active thermoregulation to maintain a stable level of thermal stress since species experiencing high thermal stress should rely on active thermoregulatory mechanisms to diminish thermal stress and prevent reaching a critical heat shock temperature during flight [20]. This pattern may also be associated with body size, as larger odonates generally have greater thermal inertia, which could help buffer temperature fluctuations [35,42,61]. In accordance with this pattern, Rocha et al. [24] described how medium- and large-sized Odonata species were favored in disturbed environments due to their more efficient thermoregulatory mechanisms. Similarly, Castillo-Pérez et al. [27] found that the most abundant species in disturbed sites within our study area exhibited a larger body size, suggesting that these species may also rely on more effective thermoregulatory mechanisms—either behavioral, such as selecting cooler microhabitats or adopting obelisk posture, or physiological, including hemolymph circulation [26,29,31,42].

While species from preserved sites experience higher thermal stress at lower maximum temperatures, their smaller body size also implies a higher rate of heat exchange through convection due to their higher surface-to-volume ratio [25]. Additionally, preserved sites have been shown to exhibit greater habitat heterogeneity and vegetation cover in previous studies [18,27], which likely provides more perching opportunities and greater access to shade. Therefore, if temperatures become extremely high, intensifying thermal stress, these species may mitigate it by seeking cooler microhabitats [25,31]. Although our statistical models accounted for interspecific variability, is some cases, our sample sizes were small for certain species, which highlights the need for long-term monitoring of thermal stress in the region to ensure more robust species-level inferences. The thermoregulatory mechanisms of species inhabiting disturbed and preserved sites require further investigation to understand their role in thermal adaptation.

Thermoregulatory mechanisms—whether physiological or behavioral, in addition to constant exposure to high temperatures—can be energetically costly [34,35,62]. In accordance with our predictions, individuals experiencing higher thermal stress—particularly in disturbed, resource-limited environments—exhibited reduced energetic condition. This pattern may also be related to their higher flight frequency [22,61], one of the most metabolically demanding activities in insects, which primarily consumes lipids and, to a lesser extent, proteins [51,63,64,65]. For instance, lipid depletion after sustained flight was observed in *Pantala flavescens* [66]. Moreover, individuals from disturbed sites exhibited lower energy reserves (proteins and lipids), regardless of their thermal stress levels. Environmental factors such as wastewater contamination and loss of vegetation cover in the study area may further contribute to this decline in energy reserves. For example, *H. americana* exhibited lower energy reserves due to increased water pH in urbanized areas [67,68], and similar patterns have been observed in dung beetles such as *Dichotomius guaribensis* under increasing urbanization [69]. These findings suggest that odonates experiencing higher thermal stress must allocate more metabolic resources to maintain homeostasis, especially under disturbed, hotter conditions. As a result, species under higher thermal stress may be more vulnerable to habitat disturbance, as they face the combined effect of environmental disturbance and the high energetic demands associated with coping with high thermal stress. This chronic demand likely accelerates the depletion of energy reserves such as lipids, which are critical for sustaining flight and reproduction.

Ultimately, a decline in energetic condition may impair fitness, as these reserves are linked to key life history traits. For example, while lipids are used to sustain flight and repel intruders during male competition over mating territories [70], proteins are used to produce eggs [71]. We hypothesize that this fitness impairment can lead to an eventual population collapse. Another important factor to consider in future studies is the time of day, as individuals under higher thermal stress might shift their activity patterns to reduce the energetic costs of coping with extreme conditions in disturbed sites. This behavioral adjustment remains to be explored. Additionally, we found weak but consistent support for a positive relationship between mean temperature and protein content, suggesting that moderate increases in ambient temperature may enhance protein availability or synthesis, potentially buffering some of the negative effects of thermal stress. This relationship should be further investigated in future studies.

We also found support for an interaction between thermal stress and site condition across the seven species that inhabit both habitat types (disturbed/preserved), partially aligning with our prediction that individuals from disturbed sites would exhibit lower thoracic mass. However, this difference was more evident in individuals experiencing high thermal stress. Our findings are consistent with those of Giménez-Gómez et al. [20], who reported that species with smaller body mass were associated with disturbed and open habitats. Our results suggest that achieving greater thoracic mass may involve higher physiological costs for individuals under high thermal stress in disturbed sites due to extreme conditions such as higher temperatures, contamination, and competition for resources [72,73,74]. In contrast, more favorable conditions in preserved sites may allow individuals with greater thoracic mass to cope with the energetic costs of maintaining a larger thoracic mass. Furthermore, it has been suggested that larger thoracic mass may be related to greater capacity for water storage in larger individuals, which could partially explain why individuals in preserved sites exhibited higher thermal stress at lower maximum temperatures. Future studies should investigate this relationship.

Odonates are widely recognized as effective bioindicators due to their sensitivity to changes in both aquatic and terrestrial habitat quality [10,75,76,77]. Numerous studies have shown that habitat degradation can lead to shifts in species composition, often resulting in assemblages dominated by resilient, generalist species capable of persisting and even increasing in abundance under suboptimal conditions [28,78,79,80]. In this context, although certain species may persist in disturbed environments, their physiological condition may be compromised. Our results support this perspective, suggesting that even resilient species may incur energetic costs under thermally stressful conditions. This pattern could also extend to other flying insect groups that experience high thermal stress, such as moths and bumblebees [21,23]. Furthermore, if maximum temperatures continue to rise at micro and macro scales, species with high thermal stress may face additional challenges. Our findings indicate that these species likely engage in behavioral or physiological thermoregulatory strategies to keep their thermal stress (i.e., the difference between body and ambient temperature) within functional limits in disturbed sites, which may lead to substantial energy expenditure. Thus, while some species remain present in disturbed habitats, their persistence comes at the cost of reduced energetic condition—highlighting a hidden vulnerability that could have long-term consequences for population viability. Thermal ecology research is essential for developing conservation strategies tailored to highly mobile insect species such as dragonflies, which play crucial roles in ecosystem functions such as transferring materials and energy from aquatic to terrestrial ecosystems [81,82] and by regulating populations of other insects, including human disease vectors such as mosquitoes [83].

## 5. Conclusions

Our study highlights how thermal stress, energy reserves, and body size vary in dragonflies inhabiting preserved and disturbed habitats within a tropical dry forest characterized by high temperatures. We found that in disturbed habitats, dragonflies maintain stable thermal stress across different temperatures, while in preserved habitats, species experience higher thermal stress at lower maximum temperatures but reduce it as temperatures increase. The energetic costs of thermoregulation appear to be higher in disturbed habitats, as individuals experiencing high thermal stress exhibited lower lipid and protein reserves. Additionally, odonate individuals with high thermal stress in preserved sites exhibited larger thoracic mass. This pattern may indicate that attaining larger thoracic mass may entail higher physiological costs under thermally stressful conditions in disturbed sites, whereas more favorable conditions in preserved sites may help offset those costs. We also found a weak but consistently supported positive association between protein content and mean ambient temperature. Although the effect was small, this result suggests a potential link between moderate temperature increases and protein availability or synthesis. Overall, our findings emphasize the importance of studying thermal ecology in insect species living in disturbed environments to inform conservation efforts.

## Figures and Tables

**Figure 1 biology-14-00956-f001:**
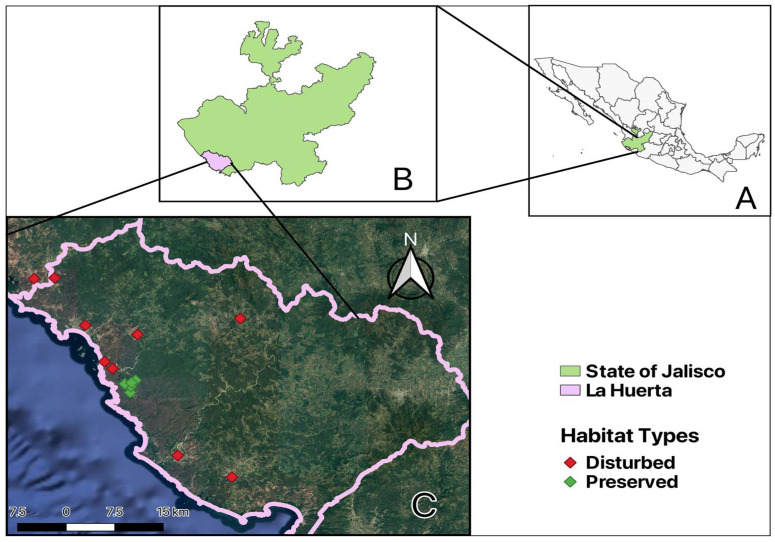
(**A**) Location of the state of Jalisco within the Mexican Republic. (**B**) Location of the municipality of La Huerta within Jalisco. (**C**) Map of sampling sites in municipality of La Huerta, Jalisco, Mexico. Symbols correspond to sampling sites.

**Figure 2 biology-14-00956-f002:**
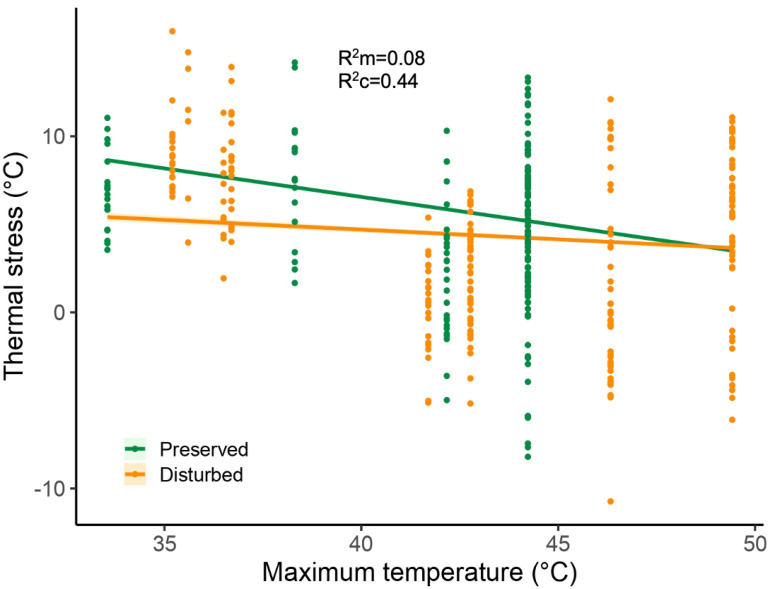
Relationship between thermal stress and maximum environmental temperature in preserved and disturbed sites for 18 species of odonates. The figure illustrates the interaction between maximum temperature and site condition (preserved/disturbed) on thermal stress, shown as regression lines with 95% confidence intervals. These lines and intervals are based on model-averaged predictions from the top competing linear mixed-effects models selected using AICc (ΔAICc < 2). Marginal (R^2^m) and conditional (R^2^c) coefficients of determination, averaged across top models, are reported to indicate the proportion of variance explained by fixed effects alone and by both fixed and random effects, respectively.

**Figure 3 biology-14-00956-f003:**
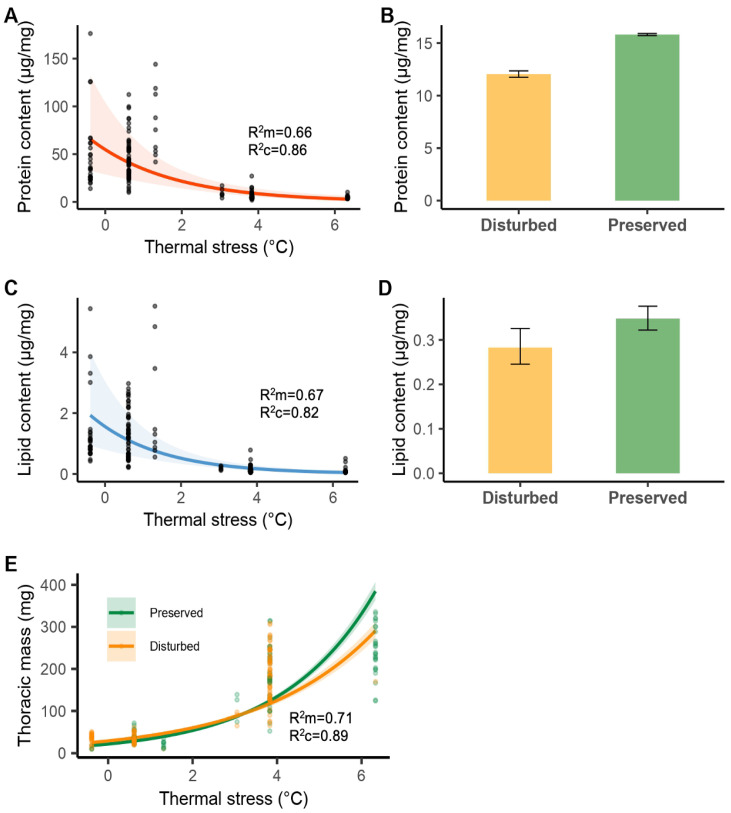
Predicted values of protein content (µg/mg), lipid content (µg/mg) and thoracic mass (mg) in relation to thermal stress (°C) and site condition (disturbed/preserved), based on model averaging across the three best-supported models (ΔAICc < 2). Panels (**A**,**C**) show predicted values along the observed gradient of thermal stress; panels (**B**,**D**) show predicted values by site condition. Shaded areas and error bars represent 95% confidence intervals. Panel (**E**) shows the interaction between thermal stress and site condition on thoracic mass, with separate slopes for disturbed and preserved sites. All values were back-transformed from the logarithmic scale for biological interpretability. Marginal (R^2^m) and conditional (R^2^c) coefficients of determination, averaged across the top models, indicate the variance explained by fixed effects alone and by both fixed and random effects, respectively.

## Data Availability

The original data presented in the study are openly available in FigShare at: https://doi.org/10.6084/m9.figshare.28953008.v1 (accessed on 7 May 2025).

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
