# Peer review of "The Physiological Cost of Being Hot: High Thermal Stress and Disturbance Decrease Energy Reserves in Dragonflies in the Wild"

_biology, 2025, doi:10.3390/biology14080956_

Round 1
Reviewer 1 Report
Comments and Suggestions for Authors
This study significantly advances understanding of insect thermal ecology by demonstrating that habitat disturbance exacerbates the physiological costs of heat stress. Authors evaluated thermal stress in 16 insect dragonfly species during two sampling periods in preserved and disturbed sites within a tropical dry forest in western Mexico, and compared energetic condition (lipid and protein content) and thoracic mass for the seven most abundant species between both habitat types. The results shown that Thermal stress and body size were positively associated, and both were linked to lower lipid and protein content. The work is very interesting. The review is enjoying. There are some comments as below.
Major
1. The microclimatic condition changed maybe more important alters the insect abundance and diversity. I suggest authors should more introduce those. Temparature increase linked to declines in insect, it should be linke the region. The tolerance capacity is very improtant to face the temperature including higher or lower, please check.
2. please introduce the more information about the dragolyies, including thermal physiology, thermoregulatory including behavioral or physiological, energy cost.
3. compared with two sampling periods. The two disturbed sites near mountain, the microhabit is different with preserved, why selected the two sites.
4. the relationship between body size and thermal stress, I suggest author reanalysis it, select the site (disturbed and preserved) as the type. The thermal stress variation in disturbed is lower than that in preserved, I cannt understand. I thinks the envirnmental varaition in preserved is higher than that in diturbed. Please check.
Minor:
Please check the species and geuns name.
Author Response
Author's Reply to the Review Report (Reviewer 1)
Summary
Thank you very much for your valuable comments. We have carefully considered each of your suggestions to substantially improve our manuscript. Below, we provide a detailed response to each point.
Comments 1. The microclimatic condition changed maybe more important alters the insect abundance and diversity. I suggest authors should more introduce those. Temparature increase linked to declines in insect, it should be linke the region. The tolerance capacity is very improtant to face the temperature including higher or lower, please check.
Response 1. We sincerely thank the reviewer for this insightful comment. We fully agree that microclimatic conditions are critical drivers of insect abundance and diversity, particularly for ectothermic organisms. Although few studies have explicitly documented a direct link between rising temperatures and insect declines in our specific region, we have expanded the introduction to better address how microclimatic variation—particularly temperature changes associated with vegetation loss—can influence insect populations. In the revised manuscript, we now include both global and local examples, including observations from our own study region, that highlight how microclimatic shifts can affect insect thermal tolerance. These additions appear in Lines 55–65.
Comments 2. please introduce the more information about the dragonflies, including thermal physiology, thermoregulatory including behavioral or physiological, energy cost.
Response 2. We appreciate the reviewer’s suggestion to expand the information on dragonflies’ thermal biology. In response, we have included more detailed description of both behavioral and physiological thermoregulatory mechanisms in Anisoptera. To the best of our knowledge, this is the first study to explicitly evaluate the energetic condition of Anisoptera adult species in the context of thermal stress in warm and disturbed environments. These additions appear in Lines 83-104 of the revised manuscript.
Comments 3. compared with two sampling periods. The two disturbed sites near mountain, the microhabit is different with preserved, why selected the two sites.
Response 3. We thank the reviewer for this observation. We included two sampling periods to increase the robustness of our study, and in both cases, data were collected using consistent methods. The 2019 data correspond to previously published measurements of body temperature with the same thermographic camera unit; more details have been added in Lines 172–180. These data were incorporated into the present analysis to strengthen the temporal scope of the study. Sampling year was included in the model to account for potential background variability between sampling periods without overparameterizing the model.
The two disturbed sites mentioned are not located closer to the mountain; rather, they are situated in crop fields near a road. We selected them because they belong to the tropical dry forest ecosystem and represent disturbed conditions within the same ecological context. While disturbance has altered their microclimatic conditions, these sites are ecologically comparable to preserved sites in terms of vegetation type and broader landscape characteristics. Additionally, potential variation associated with sampling sites was accounted for in the models by including site as a random effect.
Comments 4. the relationship between body size and thermal stress, I suggest author reanalysis it, select the site (disturbed and preserved) as the type. The thermal stress variation in disturbed is lower than that in preserved, I cannt understand. I think the envirnmental varaition in preserved is higher than that in diturbed. Please check.
Response 4. We thank the reviewer for this comment. In our analysis, we assessed the effects of maximum temperature, site condition (preserved vs. disturbed), and body size (total length) on thermal stress using a linear mixed-effects model. The interaction between maximum temperature and site condition (preserved vs. disturbed) was included to explicitly evaluate whether the response of thermal stress to maximum temperature varied between preserved and disturbed sites. Our results showed that thermal stress decreased with increasing maximum temperature (β = −0.32, p < 0.001), but this relationship was significantly moderated by site condition (interaction term: β = 0.21, p = 0.041). This indicates that the slope of thermal stress in response to maximum temperature differed between preserved and disturbed sites. Specifically, individuals in disturbed sites showed a flatter relationship, which we hypothesized may be due to more stable and consistently warmer microclimatic conditions resulting from vegetation loss. In contrast, preserved sites may present greater microclimatic heterogeneity (e.g., shade vs sun patches), leading to greater variability in thermal stress responses. This hypothesis has been added to the discussion section Lines 355–365.
Minor:
Please check the species and geuns name.
Response: Thank you for this observation. We have reviewed and corrected the species and genus names as needed.
Reviewer 2 Report
Comments and Suggestions for Authors
Dear authors,
I found your paper interesting, and it might be published, but some details need clarification:
Thermal stress measurements are the basis for this paper, but how you measured environmental temperatures is unclear. This is very important since the data you are presenting can be biased in several ways depending on how the environmental temperatures were measured.
Looking at the species communities in the preserved and disturbed habitats, their composition is similar, sharing several species. Also, all the species in your study are wide-ranging species. This shows that despite the vegetation coverage having been conserved in the past decades, the whole community is generalist, there are no endemic or specialist species, probably due to the harsh conditions in the region. So the comparisons between your habitats should be carefully made.
Finally, you are citing your own work most of the time, ignoring other studies in other regions which, takes you to make statements as: "Odonates are considered highly resilient to habitat modification although the supporting evidence has used presence/absence of individuals and/or species for this claim". This is false in other regions, odonates are considered great habitat bioindicators due to their sensitivity to habitat change. Please, read more and compare with odonate communities in other areas.
Attached is the manuscript with some other comments.
I hope this will help you improve the manuscript for a future publication.

Author Response
Author's Reply to the Review Report (Reviewer 2)
Summary
Thank you for your comments and suggestions. We have carefully addressed each point to substantially improve the manuscript. Below, we provide a detailed response to each point.
Comments 1. Recent publications on dragonflies and other insects from the Neotropics have shown this, please cite them.
Response 1. Thank you for your valuable comment. We have now cited examples from recent studies on insects in Neotropics, including dragonflies (line 57).
Comments 2. What is the difference in temperature between preserved forest and disturbed habitats?
Response 2. Thank you for your comment. We addressed this difference in the Results sections and included the maximum temperatures values as well (lines 277–282)
Comments 3. What is the forest area there?
Response 3. Thank you for your comment. In this revised version, we have included the area of the Chamela-Cuixmala Reserve (protected area), as well as the surrounding non-protected area where disturbed sites are located (lines 148–150).
Comments 4. Panel B shows Jalisco State, but this state is not shown on panel A or C, you should showcase the sampling area region. Panel C has a similar problem, it does not bring a lot of information, where is it related to panel A, what is the information you want to
Response 4. Thank you for your valuable comments. We have revised the map to improve clarity. In this new version, we clearly show the State of Jalisco and have outlined the study area (Municipality of La Huerta). We also improved the visual connection between panels A, B and C to better contextualize the location of the sampling region.
Comments 5. Do you know the precision of this instrument?
Response 5. Thank you for your observation. In response, we have added the measurement accuracy of the thermal camera model FLIR E6 used in our study to the revised manuscript (Line 187).
Comments 6. Please, explain it briefly here.
Response 6. Thank you for your suggestion. In this revised version, we have briefly explained the method as requested (Lines 202–205).
Comments 7. Why are you using a generalized model if you are going to pick the Gaussian distribution? It will be a normal linear mixed model as the one described in the above paragraph or not?
Response 7. Thank you for your observation. Although both models (GLMM and linear mixed-effects) were fitted using a Gaussian error structure, we chose to present the generalized linear mixed model (GLMM) using the glmmTMB function because the linear mixed model (lme) violated the assumption of homoscedasticity, as shown by a significant heteroscedasticity (p < 0.001), even after log- transforming the response variable to improve normality and variance stabilization. In contrast, the GLMM met all key assumptions: the Breusch-Pagan test indicated no heteroscedasticity (p= 0.20), and residual diagnostics using the DHARMa package supported the model´s adequacy (uniformity: p= 0.078; dispersion: p= 0.688). While the GLMM used a Gaussian distribution– similar to a standard linear model– glmmTMB provided a more flexible and diagnostically robust framework, which is why we retained it in the final analysis.
Comments 8. Are you presenting the mean temperatures or the maximum temperatures?
Response 8. Thank you for your comment. In this revised version, we clarify that both mean and maximum temperatures are presented.
Comments 9. You are presenting the mean environmental tempertures, but you are using the maximum environmental temperatures in the analysis, why?
Response 9. Thank you for your comment. In this revised version, we have clarified in the Statistical analyses section why maximum temperature was used in some analyses. Specifically, maximum temperature improved model performance and met model assumptions in the linear mixed models assessing thermal stress (Lines 252–253; table S3), whereas mean temperature was more appropriate for the generalized linear mixed models exploring energetic reserves (Lines 264–265; table S3).
Comments 10. You need to define this clearly, what is a lower maximum temperature?
Response 10. Thank you for your comment. In this case by “lower maximum temperature” we refer to the lowest value among all the maximum temperatures recorded across sites– that is, the minimum value within the set of maximum temperature observations. This clarification has been added in Lines 290–293 of the revised manuscript.
Comments 11. There are a lot of studies claiming odonates sensitivity to habitat modification, that is why they are considered great bioindicators. Please rethink this, and discuss in the light of most findings, not only geographically biased communities.
Response 11. Thank you very much for this thoughtful comment. We agree that odonates are widely recognized as reliable bioindicators due to their sensitivity to both aquatic and terrestrial habitat conditions. We have adjusted the discussion (Lines 413–417) to reflect this broader understanding, while still highlighting that, according to our results, species persisting in disturbed habitats may incur energetic costs.
Comments 12. Looking at the species communities in the preserved and disturbed habitats, their composition is similar, sharing several species. Also, all the species in your study are wide-ranging species. This shows that despite the vegetation coverage having been conserved in the past decades, the whole community is generalist, there are no endemic or specialist species, probably due to the harsh conditions in the region. So the comparisons between your habitats should be carefully made.
Response 12. Thank you for the comment. Although the species are mostly generalists and shared between habitats, our results (see Discussion, lines 413–433) show that individuals in disturbed habitats exhibit reduced energy reserves. This supports the ecological relevance of preserved vs disturbed habitat distinction, as it reflects clear differences in physiological condition despite community-level similarities.
Reviewer 3 Report
Comments and Suggestions for Authors
The manuscript is well suited for this special issue and the question is clearly presented. I like the ideas and the hypotheses. However, at present, the data could be interpreted and analyzed in several ways and unfortunately, the methodology is currently lacking information to be convinced that the data is being interpreted and analyzed correctly (please see below - methods and results).
The introduction is nice and goes from the broad to the specifics, but does so a bit unorganized. It could benefit from a more detailed and organized explanation of important terms and a better organization to improve the flow of ideas. In that way, it will be easier to understand the logic of the study all the way to the specific question. For instance, lines 92-96 present important information that should be seen by the reader before lines 84-92. This change should help the reader follow the logic of the study more easily.
The predictions seem to include an interaction term that is not specified in the last paragraph of the Introduction. The authors are only expecting poorer energetic reserves in individuals with higher thoracic mass (only larger individuals because they are under higher thermal stress). This effect is not expected in smaller individuals, so it is confusing to see the predictions as currently stated. This nuance should be detailed. Perhaps better to change the third prediction as the second and the second as the third (making this clarification).
It appears the study was done during the rainy season and not during the hottest months of the dry season (in the conclusion is says that the tropical dry forest where the study was done is characterized by high temperatures).
Paragraph two introduces important concepts (thermal stress, heliothermic, thermoconforming), but the difference between these terms is not clearly explained. For instance, it seems that a heliothermic species can also be thermoconforming (i.e., ectothermic) and has a lower thermal stress. It is important that the reader know the equivalency and/or difference between terms explained here so that s(he) doesn't get confused later on. Thermal stress is a particularly important concept because it is used throughout the manuscript. Hence, this term should be introduced in its own right, rather than tangentially as part of Gimenez-Gomez's study. Also, this term is introduced again in lines 82-83, so perhaps best to move those lines into this second paragraph. Similarly, in lines 85-86, the link between thermal stress and metabolic rate is not clearly explained. This should be explained in the third paragraph, where you give mechanisms for the consequences of high temperature for insect biodiversity decline (e.g., death of fat body cells). At the moment, the author arrives at lines 85-86 without having heard about the relationship between metabolism and heat stress, so it is difficult to make sense of the information present in that paragraph.
line 55-56 - the link between insects 'depend(ing) on environmental temperature' and change in microclimatic conditions is somewhat unclear. It reads as if it could go either way - becoming cooler or becoming hotter, but the anthropogenic effects the authors are talking about here (removal of forest cover) lead to hotter conditions. Hence, this sentence could be more specific and particularly reference increasing temperatures (not decreasing) in these microclimates.
Methods/Results: An accurate description of the disturbed vs preserved sites is needed. These sites could differ in a myriad of other conditions besides temperature and canopy cover, for instance, in stream width, current, or whether they represent ponds, rivers of streams. In an extreme case, the disturbed sites could be all ponds and the preserved sites all rivers and this in itself will produce different species compositions/body conditions in disturbed vs preserved sites that can be attributed not to the disturbance per se but to the habitat. Before attrributing any effects to temperature, we need to know how comparable are the disturbed vs undisturbed sites in: habitat type (stream or pond), openness, stream width and current. Or at least be convinces that all sites are similar in these respects.
Secondly, it is important to relate the datalogger data to a wider date range. It seems that for some sites, ambient temperature was only recorded during 2 days. It is important to note the number of days that dataloggers were deployed in each site (this can be done in a table). Please clarify this (the duration of the temperature recording) and relate it to a wider dataset to convince the reader that your measurements reflect the normal overall conditions. Can you access a public climate dataset for these regions? A stronger analyses would relate the datalogger data with a climatic data set spanning more days.
Most importantly, time of day should be included in the analysis. Temperature varies with time of day and some damselfly species could have been captured in the morning and some after midday - this would create very strong differences in the thermal stress that are due not to species but to time of day. This risk is exacerbated given that the sample size for species is low. For instance, if n = 2 for a species that has low thermal stress (see table for sample sizes for species and figure of thermal stress), how can we be convinced that this species has low thermal stress rather than having simply being cut at 10am or 3pm vs 12 midday? Similarly, how can we be convinced that species with high thermal stress weren't simply species that were caught at midday? Having sampled between 10am and 3pm and low sample sizes makes this a big concern. Finally, FigureS1 should be also displaying a measure of variation along the mean for each species. Sample sizes above each bar are also encouraged.
In the methods, it is important to mention for how long the damselflies were kept in the 5 °C cooler before being placed in the freezer. Otherwise, future studies won't be able to compare their findings with these ones. Also, it is important to know this because individuals captured in the morning needed to wait longer periods in the cooler and this could affect their fat reserves - please explain how this can affect results and whether or not it can be an important concern.
Thermal stress should, ideally, be compared in the same species in different habitats (disturbed vs preserved). I guess this is not always possible (or maybe never) given different microclimatic preferences of species.
The protocol for extracting lipid and protein content (Foray et al) needs to be at least briefly explained here so the reader is not forced to search for Foray et al to understand this manuscript.
For model selection, please include a table of the models that were evaluated along with their AIC values. Otherwise the reader simply needs to trust the choice made by the authors because the other models were not included.
minor comments:
line 58 - 'local declines' or 'global declines'?
line 64 - comma after 'exposure' is missing.
line 65 - The point here is unclear. Heliothermic insects are also thermoconforming if one takes a broad definition of 'thermoconforming' (i.e., ectothermic). Please explain this finding from Rocha et al more clearly.
line 77 - perhaps substitute 'In relation to this' by 'For instance'
line 78 - 'death' instead of 'dead' in two instances
line 83 - there is an inaccurate piece of the sentence - it is mentioned that thermal stress is primarily influenced by muscular activity, but environmental temperature is probably more important, no? Perhaps the confusion arises from using 'it'. 'It' refers to thermal stress, but I think the authors are referring to body temperature.
line 88 - 'smaller' instead of 'small'
lines 92-93 - it is mentioned that ' habitat disturbance can also influence prey availability and 93 nutritional quality'. Please be more specific. Influence makes reference to being higher or lower. I am guessing the authors mean lower prey availability and quality? This needs to be clearly explained so that the reader can understand the logic of the question.
line 101 - 'predicted'
line 114 - please mention the range of temperatures for both the dry and wet seasons.
line 117 - 'nine' instead of 'night'
line 128 - 'sampling' instead of 'sample'
liine 131 - ambient and maximum? Do you mean average and maximum ambient temperatures?
line 159 - the rational for only using males should be given in the Introduction rather than this far down into the manuscript. The predictions that apply should be made specific to males, then.
lines 253-255 - It is necessary to show the data graphs (or other data description) for the comparisons mentioned here: size and mass. Only the stats are shown but it is necessary to see the graphed or summarized data.
line 294 - please explain what the obelisk posture produces, and what are the changes in hemolymph circulation.
In the Discussion, the results about energy efficiency or metabolism are only very briefly mentioned in lines 272-278. Please explain what these results mean in terms of metabolism. costs, and the general question and context of your study.
line 342 - please explain what the other studies in odonates have found. It is weird that the predictions did not take these other odonate studies into account. Why is this info only used in the Discussion and not the Introduction.
line 369 - 'despite having lower energy reserves' can be deleted because it does not make much sense after reading the next sentence. The next sentence mentions that larger body sizes have higher thermal tolerances.
Author Response
Author's Reply to the Review Report (Reviewer 3)
Summary
We sincerely thank you for your constructive feedback. Your comments helped us significantly improve the manuscript. In this revised version, we have clarified key concepts, strengthened the methodological descriptions, and provided additional data and tables to support our interpretations. All your suggestions have been addressed point by point below.
Comments 1: The introduction is nice and goes from the broad to the specifics but does so a bit unorganized. It could benefit from a more detailed and organized explanation of important terms and a better organization to improve the flow of ideas. In that way, it will be easier to understand the logic of the study all the way to the specific question. For instance, lines 92-96 present important information that should be seen by the reader before lines 84-92. This change should help the reader follow the logic of the study more easily.
Response 1: Thank you for the suggestion. We have reorganized the introduction to improve the logical flow. Although the specific content and line order have changed due to adjustments based on multiple reviewer suggestions, the ideas mentioned (formerly lines 92–96) are now integrated earlier in the introduction (current lines 100–110).
Comments 2: The predictions seem to include an interaction term that is not specified in the last paragraph of the Introduction. The authors are only expecting poorer energetic reserves in individuals with higher thoracic mass (only larger individuals because they are under higher thermal stress). This effect is not expected in smaller individuals, so it is confusing to see the predictions as currently stated. This nuance should be detailed. Perhaps better to change the third prediction as the second and the second as the third (making this clarification).
Response 2: Thank you for the suggestion. We revised the predictions for clarity and restructured their order as recommended. The third prediction now explicitly reflects that poorer energetic reserves are expected mainly in larger individuals under high thermal stress.
Comments 3: It appears the study was done during the rainy season and not during the hottest months of the dry season (in the conclusion is says that the tropical dry forest where the study was done is characterized by high temperatures).
Response 3: Thank you for your observation. Indeed, the sampling was conducted during the rainy season, as this is the only period of the year when dragonflies are sufficiently abundant for ecological studies. However, historical climatic data for the region indicate that maximum temperatures during the rainy season are comparable to, or even higher than, those recorded during the dry season (http://www.ibiologia.unam.mx/ebchamela/www/clima.html).
Comments 4. Paragraph two introduces important concepts (thermal stress, heliothermic, thermoconforming), but the difference between these terms is not clearly explained. For instance, it seems that a heliothermic species can also be thermoconforming (i.e., ectothermic) and has a lower thermal stress. It is important that the reader know the equivalency and/or difference between terms explained here so that s(he) doesn't get confused later on. Thermal stress is a particularly important concept because it is used throughout the manuscript. Hence, this term should be introduced in its own right, rather than tangentially as part of Gimenez-Gomez's study. Also, this term is introduced again in lines 82-83, so perhaps best to move those lines into this second paragraph. Similarly, in lines 85-86, the link between thermal stress and metabolic rate is not clearly explained. This should be explained in the third paragraph, where you give mechanisms for the consequences of high temperature for insect biodiversity decline (e.g., death of fat body cells). At the moment, the author arrives at lines 85-86 without having heard about the relationship between metabolism and heat stress, so it is difficult to make sense of the information present in that paragraph.
Response 4. We thank the reviewer for this insightful comment. In the revised version, we have clarified the differences between thermal stress, heliothermic, and thermoconforming to avoid confusion. We now introduce the concept of thermal stress explicitly and independently of other studies. Additionally, we reorganized the relevant content to place this definition earlier in the paragraph, as suggested. We have also clarified the link between thermal stress and metabolic rate, highlighting how higher body temperatures can elevate energetic costs. These changes aim to improve conceptual clarity and logical flow throughout the introduction.
Comments 5. line 55-56 - the link between insects 'depend(ing) on environmental temperature' and change in microclimatic conditions is somewhat unclear. It reads as if it could go either way - becoming cooler or becoming hotter, but the anthropogenic effects the authors are talking about here (removal of forest cover) lead to hotter conditions. Hence, this sentence could be more specific and particularly reference increasing temperatures (not decreasing) in these microclimates.
Response 5. Thank you for your comment. In the revised version, we have made the microclimatic consequences of vegetation removal more explicit by emphasizing that these changes typically lead to increased temperatures and reduced humidity (Lines 53–65).
Comments 6. Methods/Results: An accurate description of the disturbed vs preserved sites is needed. These sites could differ in a myriad of other conditions besides temperature and canopy cover, for instance, in stream width, current, or whether they represent ponds, rivers of streams. In an extreme case, the disturbed sites could be all ponds and the preserved sites all rivers and this in itself will produce different species compositions/body conditions in disturbed vs preserved sites that can be attributed not to the disturbance per se but to the habitat. Before attrributing any effects to temperature, we need to know how comparable are the disturbed vs undisturbed sites in: habitat type (stream or pond), openness, stream width and current. Or at least be convinces that all sites are similar in these respects.
Response 6. Thank you very much for this valuable observation. Both preserved and disturbed sites are located within the same tropical dry forest ecosystem. Within each category (Preserved or disturbed), we sampled a variety of water bodies, including permanent and temporal rivers, lakes, and ponds. To clarify this, we have added a supplementary table detailing the habitat type (e.g., stream and pond: Table S1), water permanence, and other relevant site characteristics. Furthermore, we accounted for variability among sites by including “site” as a random factor in our statistical models, which helps control for unmeasured environmental differences.
Comments 7. Secondly, it is important to relate the datalogger data to a wider date range. It seems that for some sites, ambient temperature was only recorded during 2 days. It is important to note the number of days that dataloggers were deployed in each site (this can be done in a table). Please clarify this (the duration of the temperature recording) and relate it to a wider dataset to convince the reader that your measurements reflect the normal overall conditions. Can you access a public climate dataset for these regions? A stronger analyses would relate the datalogger data with a climatic data set spanning more days.
Response 7. Thank you for your thoughtful comment. Due to logistical and budgetary constraints, it was not feasible for us to deploy the dataloggers for extended periods in all the study sites. Nevertheless, most sites had temperature recordings spanning an average of five full days (non-consecutive), with a minimum of two full days. We have now specified this information more clearly in the methods section and detailed it in Table S1. Unfortunately, no public climate datasets with sufficient spatial resolution are available for our specific sampling locations. The only available data correspond to preserved areas within Chamela-Cuixmala Biosphere Reserve. However, this dataset only includes daily maximum and minimum temperatures and is limited to data collected up to 2010. Despite this limitation, the maximum temperatures we recorded in August-September 2019 and 2022 (mean of 38.95°C) are consistent with those reported in the Chamela-Cuixmala weather station for the same months (38°C in August and 39°C in September 2007; http://www.ibiologia.unam.mx/ebchamela/Tmax/tmax07.htm). We think our study remains valuable, as it incorporates microclimatic temperature measurements that provide a fine-scale perspective often lacking in broader climatic datasets. Moreover, our temperature measurements were collected during the peak of the rainy season (August-September), which coincides with the period of the highest activity and abundance of adult odonates.
Comment 8. Most importantly, time of day should be included in the analysis. Temperature varies with time of day and some damselfly species could have been captured in the morning and some after midday - this would create very strong differences in the thermal stress that are due not to species but to time of day. This risk is exacerbated given that the sample size for species is low. For instance, if n = 2 for a species that has low thermal stress (see table for sample sizes for species and figure of thermal stress), how can we be convinced that this species has low thermal stress rather than having simply being cut at 10am or 3pm vs 12 midday? Similarly, how can we be convinced that species with high thermal stress weren't simply species that were caught at midday? Having sampled between 10am and 3pm and low sample sizes makes this a big concern. Finally, FigureS1 should be also displaying a measure of variation along the mean for each species. Sample sizes above each bar are also encouraged.
Response 8. Thank you very much for this insightful and important comment. We fully understand the concern regarding the potential confounding effect of time of day on thermal stress. However, thermal stress was calculated as the difference between body temperature and ambient temperature recorded at the exact moment of capture, which inherently integrates variation associated with time of day. Thus, adding time of day as separate covariate would be redundant, as its effect already embedded in the response variable. In contrast, maximum temperature refers to the daily maximum ambient temperature recorded at each site using dataloggers and was included to characterize overall habitat-level thermal extreme conditions, not individual exposure at capture time.
Additionally, our primary research objective was to evaluate community-level patterns of thermal stress across site types (i.e., preserved vs. disturbed), rather than species-specific responses. Accordingly, we included species identity as a random effect in our mixed-effects model to account for interspecific variability and unbalanced sample sizes. Our results suggests that habitat structure (e.g., vegetation cover, shade availability) modulates thermal stress responses differently, in ways that cannot be attributed solely to time of day.
Regarding Figure S1, we agree that it may lead to misinterpretation, particularly for species with low sample sizes. While originally included for descriptive purposes, we have now decided to remove this figure entirely from the manuscript to avoid potential overinterpretation. Moreover, we now explicitly acknowledge in the revised Discussion that certain species had low n, and we highlight the need for long-term monitoring to improve inference and data representativeness (lines 361–364).
Comments 9. In the methods, it is important to mention for how long the damselflies were kept in the 5 °C cooler before being placed in the freezer. Otherwise, future studies won't be able to compare their findings with these ones. Also, it is important to know this because individuals captured in the morning needed to wait longer periods in the cooler and this could affect their fat reserves - please explain how this can affect results and whether or not it can be an important concern.
Response 9. Thank you for your comment. The maximum time between capture and placement in the freezer was five hours. However, a temperature of 5°C is sufficiently low to significantly reduce the metabolic rate of dragonflies, preventing further activity. Additionally, we deliberately excluded carbohydrate measurements from our analyses, as these are the most rapidly depleted energy reserves. Lipids, in particular, have a slower turnover rate in insects and are less sensitive to short-term post-capture (Arrese & Soulages, 2010; Li et al., 2023), reducing the likelihood of significant depletion during the holding period. Therefore, we consider that the delay between capture and freezing did not significantly affect the lipid or protein reserves measured in our study. This information has now been explicitly clarified in the Methods section.
References
Arrese, E. L., & Soulages, J. L. (2010). Insect fat body: energy, metabolism, and regulation. Annual Review of Entomology, 55(1), 207–225.
Li, X., Zhou, Y., & Wu, K. (2023). Biological Characteristics and Energy Metabolism of Migrating Insects. In Metabolites(Vol. 13, Issue 3). MDPI. https://doi.org/10.3390/metabo13030439
Comment 10. Thermal stress should, ideally, be compared in the same species in different habitats (disturbed vs preserved). I guess this is not always possible (or maybe never) given different microclimatic preferences of species.
Response 10: Thank you for your comment. Unfortunately, it was not always possible to compare thermal stress within the same species across both disturbed and preserved habitats, despite extensive sampling efforts. This limitation is likely due to species-specific microclimatic preferences and variation in local resource availability, which may restrict the occurrence of certain species to particular habitat types.
Comment 11. The protocol for extracting lipid and protein content (Foray et al) needs to be at least briefly explained here so the reader is not forced to search for Foray et al to understand this manuscript.
Response 11. Thank you for your comment. We have added a brief explanation of the lipid and protein extraction protocol to improve clarity for the reader.
Comment 12. For model selection, please include a table of the models that were evaluated along with their AIC values. Otherwise the reader simply needs to trust the choice made by the authors because the other models were not included.
Response 12. Thank you for your suggestion. We have now included the requested table listing the best-supported models based on AIC in the supplementary material (Table S3).
Minor comments:
Comment 13. 'local declines' or 'global declines'?
Response 13. Thank you for your observation. We have clarified this point by specifying “global declines” in line 66 of the revised version.
Comment 14. line 64 - comma after 'exposure' is missing.
Response 14. Thank you for your observation. In this revised version, we rephrased the sentence, so the structure has changed (now located in lines 78–82).
Comment 15. line 65 - The point here is unclear. Heliothermic insects are also thermoconforming if one takes a broad definition of 'thermoconforming' (i.e., ectothermic). Please explain this finding from Rocha et al more clearly.
Response 15. Thank you for your comment. It is correct that heliothermic insects are ectotherms. However, according to classifications such as those proposed by May (1979) and Sanborn (2005), heliotherms are ectotherms that exhibit a degree of behavioral thermoregulation through sun exposure. As noted by Rocha et al. (2023) “heliothermic odonates have a larger body and, consequently, lower thermal conductance, and their activities are determined mainly by solar irradiation.” In the revised version (lines 78–82), we have clarified this distinction and more clearly explained the findings of Rocha et al.
May, M. (1979). Insect Thermoregulation. Annual Review of Entomology, 24, 313–349. https://doi.org/10.1146/annurev.en.24.010179.001525
Rocha, T. S., Calvão, L. B., Juen, L., & Oliveira-Junior, J. M. B. (2023). Effect of environmental integrity on the functional composition of the Odonata (Insecta) community in streams in the Eastern Amazon. Frontiers in Ecology and Evolution, 11. https://www.frontiersin.org/journals/ecology-and-evolution/articles/10.3389/fevo.2023.1166057
Sanborn, A. (2005). Thermoregulation in Insects BT - Encyclopedia of Entomology. In Encyclopedia of Entomology (pp. 2224–2225). Springer Netherlands. https://doi.org/10.1007/0-306-48380-7_4289
Comment 16. line 77 - perhaps substitute 'In relation to this' by 'For instance'
Response 16. Thank you for your suggestion. We have implemented the recommended change in the revised version of the manuscript.
Comment 17. line 78 - 'death' instead of 'dead' in two instances
Response 17. Thank you for your observation. We have corrected the two instances in lines 108–110.
Comment 18. line 83 - there is an inaccurate piece of the sentence - it is mentioned that thermal stress is primarily influenced by muscular activity, but environmental temperature is probably more important, no? Perhaps the confusion arises from using 'it'. 'It' refers to thermal stress, but I think the authors are referring to body temperature.
Response 18. Thank you for your observation. We have revised this section accordingly, and the updated information can now be found in lines 70–74 of the manuscript.
Comment 19. line 88 - 'smaller' instead of 'small'
Response. Thank you for your correction. We have replaced the word accordingly in this revised version.
Comment 19. lines 92-93 - it is mentioned that ' habitat disturbance can also influence prey availability and 93 nutritional quality'. Please be more specific. Influence makes reference to being higher or lower. I am guessing the authors mean lower prey availability and quality? This needs to be clearly explained so that the reader can understand the logic of the question.
Response 19. Thank you for your suggestion. We have clarified that habitat disturbance can reduce prey availability and lower nutritional quality in lines 113–115.
Comment 20. line 101 - 'predicted'
Response 20. Thank you for your suggestion. We have corrected the term as recommended in the revised version.
Comment 21. line 114 - please mention the range of temperatures for both the dry and wet seasons.
Response 21. Thank you for your observation. We have now specified the temperature ranges for both the dry and rainy seasons in the study area, as suggested (lines 151—154).
Comment 22. line 117 - 'nine' instead of 'night'
Response 22. Thank you for pointing this out. We have corrected the typographical error in the revised version.
Comment 23. line 128 - 'sampling' instead of 'sample'
Response 23. Thank you for your observation. The correction has been made in the revised version.
Comment 24. line 131 - ambient and maximum? Do you mean average and maximum ambient temperatures?
Response 24. Thank you for your observation. We have clarified the terminology in this revised version (lines 172—180).
Comment 25. line 159 - the rational for only using males should be given in the Introduction rather than this far down into the manuscript. The predictions that apply should be made specific to males, then.
Response 25. Thank you for your suggestion. We did not include females because they are difficult to locate and identify in the field. In this revised version, we have clarified in the introduction that our predictions specifically apply to males (lines 133—134).
Comment 26. lines 253-255 - It is necessary to show the data graphs (or other data description) for the comparisons mentioned here: size and mass. Only the stats are shown but it is necessary to see the graphed or summarized data.
Response 26. Thank you for your observation. We have now included the corresponding figure that visually represents the results (Figure 3E and 3F).
Comment 27. line 294 - please explain what the obelisk posture produces, and what are the changes in hemolymph circulation.
Response 27. Thank you for your observation. We have now provided a more detailed description of the obelisk posture and we described the changes in hemolymph circulation (lines 92—100).
Comment 28. In the Discussion, the results about energy efficiency or metabolism are only very briefly mentioned in lines 272-278. Please explain what these results mean in terms of metabolism. costs, and the general question and context of your study.
Response 28. Thank you for your suggestion. We have expanded this discussion on energy reserves and metabolic costs in revised manuscript to better connect these results with the broader context of our study. These changes are reflected in lines 378—386.
Comment 29. line 342 - please explain what the other studies in odonates have found. It is weird that the predictions did not take these other odonate studies into account. Why is this info only used in the Discussion and not the Introduction.
Response 29. Thank you for your observation. We have clarified this point by integrating the relevant findings from previous studies into the introduction (lines 124—129).
Comment 30. line 369 - 'despite having lower energy reserves' can be deleted because it does not make much sense after reading the next sentence. The next sentence mentions that larger body sizes have higher thermal tolerances
Response 30. Thank you for your suggestion. We have removed the phrase to improve clarity and avoid redundancy.
Round 2
Reviewer 2 Report
Comments and Suggestions for Authors
There are two Statistical paradigms mixed in the analysis: information theory and hypothesis testing. This doesn´t make any sense. If you are going to use multimode selection based on AIC scores, then you have to compare it with the null model; you cannot calculate p-values, which is a completely different approach. If you want to calculate p-values, then you don´t make the multimodel selection.
Please correct this. You are making a bad mistake by mixing two different statistical approaches.
Also, you did not answer other simple questions on the Analysis, like why do you generalize a Gaussian distributed model?
Finally, you keep ignoring literature from the tropical regions on thermal odonate thermophysiology.
Author Response
Author's Reply to the Review Report (Reviewer 2)
Summary
Thank you for your comments and suggestions. In this revised version, we have adopted a single statistical paradigm for our analyses and included additional examples of thermal ecology and ecophysiology in tropical Odonata. We have carefully addressed each point to substantially improve the manuscript. Below, we provide a detailed response to each point.
Comments 1. There are two Statistical paradigms mixed in the analysis: information theory and hypothesis testing. This doesn´t make any sense. If you are going to use multimode selection based on AIC scores, then you have to compare it with the null model; you cannot calculate p-values, which is a completely different approach. If you want to calculate p-values, then you don´t make the multimodel selection. Please correct this. You are making a bad mistake by mixing two different statistical approaches.
Response 1. Thank you for this important observation. In the revised manuscript, we have removed the use of p-values and focused our inference entirely on model selection based on AIC scores. The rationale and procedures are now clearly described in the Methods section, and the results and figures have been revised to reflect this approach, using model-averaged estimates from the best-supported models (ΔAICc < 2).
Comments 2. Also, you did not answer other simple questions on the Analysis, like why do you generalize a Gaussian distributed model?
Response 2. Thank you for this observation. In the revised manuscript, we no longer use generalized linear models and instead apply linear mixed-effects models (LMMs). These changes are now described in the methods section.
Comments 3. Finally, you keep ignoring literature from the tropical regions on thermal odonate thermophysiology.
Response 3. Thank you for this important observation. In the revised version, we have incorporated and explicitly discussed additional literature from tropical regions addressing thermal ecology in Odonata, both at the community and physiological levels. For example, changes in Odonata community composition in response to canopy loss and increased solar radiation have been documented in tropical forests (e.g. references 12-13). Additionally, we now refer to Bota-Sierra, et al. [17], who demonstrated that thermal tolerance of tropical damselflies and dragonflies is closely linked to microclimatic conditions. Similarly, research conducted in Mexican tropical dry forests [18] found that Odonata populations inhabiting disturbed, warmer sites exhibited increased upper thermal limits, suggesting a physiological adjustment to higher temperatures in disturbed sites. Together, these studies underscore the importance of integrating physiological and ecological perspectives to understand how tropical insects respond to microclimatic changes.
Reviewer 3 Report
Comments and Suggestions for Authors
The authors submitted a revised version of the manuscript that incorporates most of the suggested changes. This version of the manuscript has been improved. In particular, the Introduction reads better, the key concepts have been more thoroughly explained, and the predictions are now clearer. The protocol for energetic condition is now thoroughly described. Also, the model selection is now more transparent with the authors reporting AIC values for different models in Table S3. Most importantly, the authors now commented on the site differences (disturbed vs undisturbed) in terms of their habitat characteristics (TableS1). However, habitat differences should still be discussed as a potential confounding variable (or arguments against this possibility should be clearly stated in the manuscript). For instance, the 'seasonal pond' habitat is present in half the preserved sites but in none of the disturbed sites. Can this affect the results obtained or their interpretation? Another important aspect in which sites could differ is in prey availability (see lines 113-116). How can we be sure that the energetic differences were not produced by diet rather than thermal stress or coping mechanisms for thermal stress? I agree it is useful to include site as a random factor to account for some of this variability, but I did not see a random factor structure in the model selection table (table S3). The most important caveat of the paper, in my opinion, has not been fully addressed. As stated before, time of day should be included in these sorts of comparisons because it is not the same to measure thermal stress at a preserved site at 9 am and compare it with a measurement taken at a disturbed site at 1 pm. Unless the capture times were kept evenly distributed among both site-categories (i.e., an equal number of captures at 9am, 10am, etc), then time of day is likely to add much noise to the data, which at present is not accounted for. Due to these two caveats, I don't think that it is possible at this stage to attribute the differences found solely to temperature conditions differing at the sites. These caveats of the study should be commented on in the Discussion.
Author Response
Author's Reply to the Review Report (Reviewer 3)
Summary
We sincerely thank you for your constructive feedback. Your comments helped us significantly improve the manuscript. In this revised version, we clarified why site identity was included as a random factor in some models but not in others, and we justified our decision not to include time of day as a fixed effect—while acknowledging that it may still be a variable worth exploring in future studies.
Comments 1: The authors submitted a revised version of the manuscript that incorporates most of the suggested changes. This version of the manuscript has been improved. In particular, the Introduction reads better, the key concepts have been more thoroughly explained, and the predictions are now clearer. The protocol for energetic condition is now thoroughly described. Also, the model selection is now more transparent with the authors reporting AIC values for different models in Table S3. Most importantly, the authors now commented on the site differences (disturbed vs undisturbed) in terms of their habitat characteristics (TableS1). However, habitat differences should still be discussed as a potential confounding variable (or arguments against this possibility should be clearly stated in the manuscript). For instance, the 'seasonal pond' habitat is present in half the preserved sites but in none of the disturbed sites. Can this affect the results obtained or their interpretation?
Response 1. Thank you very much for your valuable comments and for acknowledging the improvements in the revised manuscript. We sincerely apologize for the error in the classification of habitat types; it was due to a translation oversight. In fact, the majority of the sampling sites, both preserved and disturbed were seasonal ponds, which is typical of this tropical dry forest. This has been corrected in the updated version of Table S1.
Regarding the concern about habitat heterogeneity as a potential confounding factor, we carefully evaluated the inclusion of site identity as random effect in alternative models. However, model selection based in AIC consistently indicated that models including site or pond type did not improve fit and received less support compared to the top-ranked models. As a result, site was excluded from the final models, with the exception of thoracic mass, where site identity was retained as a random effect due to its stronger relative support. We have clarified these points in the revised methods and discussion sections of the manuscript.
Comments 2. Another important aspect in which sites could differ is in prey availability (see lines 113-116). How can we be sure that the energetic differences were not produced by diet rather than thermal stress or coping mechanisms for thermal stress? I agree it is useful to include site as a random factor to account for some of this variability, but I did not see a random factor structure in the model selection table (table S3). The most important caveat of the paper, in my opinion, has not been fully addressed. As stated before, time of day should be included in these sorts of comparisons because it is not the same to measure thermal stress at a preserved site at 9 am and compare it with a measurement taken at a disturbed site at 1 pm. Unless the capture times were kept evenly distributed among both site-categories (i.e., an equal number of captures at 9am, 10am, etc), then time of day is likely to add much noise to the data, which at present is not accounted for. Due to these two caveats, I don't think that it is possible at this stage to attribute the differences found solely to temperature conditions differing at the sites. These caveats of the study should be commented on in the Discussion.
Response 2. Thank you for your thoughtful comments. We agree we cannot fully rule out the possibility that the observed differences between sites in energetic condition may be influenced by factors other than thermal stress, including prey availability. However, our model selection results consistently indicated that thermal stress was negatively associated with protein and lipid content and was included in the most strongly supported models. In contrast, site identity explained a very small proportion of the variance in protein and lipid models and was therefore excluded for parsimony. Site was retained only in the models of thoracic mass, where it received relatively stronger support. These model structure decisions are now more clearly described in the revised methods section.
Regarding time of day, we chose not to include it as a separate predictor to avoid overcomplicating the models and because of its strong relationship with ambient temperature at the time of capture. In our view, using the environmental temperature measured during sampling already captures the temporal variation in thermal exposure (e.g., De Marco Jr. & Resende, 2002; Polcyn, 1994). For example, if individuals are captured at different times of day and in different thermal conditions and sites (preserved or disturbed), those differences will be directly reflected in their thermal stress (i.e., body temperature minus ambient temperature). Furthermore, including species identity as a random factor helps to account for potential differences between daily activity patterns among species.
Additionally, we conducted exploratory analyses of the relationship between time and ambient temperature, which confirmed that temperature follows a nonlinear pattern over the day (e.g., lower values in early morning and later afternoon, and peaks at midday). As such, time of day would add complexity to the models without improving explanatory power. Nevertheless, as suggested, we now explicitly address time of capture as potential source of variation and limitations in the revised discussion section.
References
De Marco Jr., P., & Resende, D. C. (2002). Activity patterns and thermoregulation in a tropical dragonfly assemblage. Odonatologica, 31(2), 129–138.
Polcyn, D. M. (1994). Thermoregulation During Summer Activity in Mojave Desert Dragonflies (Odonata: Anisoptera). Functional Ecology, 8(4), 441–449. https://doi.org/10.2307/2390067